# Integrated Thermal Rehabilitation Care: An Intervention Study

**DOI:** 10.3390/healthcare11172384

**Published:** 2023-08-24

**Authors:** Giovanni Barassi, Maurizio Panunzio, Antonella Di Iulio, Angelo Di Iorio, Raffaello Pellegrino, Antonio Colombo, Giuseppe Di Stefano, Piero Galasso, Stefania Spina, Umberto Vincenzi, Andrea Santamato

**Affiliations:** 1Center for Physiotherapy, Rehabilitation and Re-Education-CeFiRR-Gemelli Molise, 86100 Campobasso, Italy; 2Department of Thoracic Surgery, “Santo Spirito” Civil Hospital, 65124 Pescara, Italy; 3Antalgic Mini-Invasive and Rehab-Outpatients Unit, Department of Innovative Technologies in Medicine & Dentistry, University “G. D’Annunzio”, Viale Abruzzo 322, 66100 Chieti, Italy; angelo.diiorio@unich.it; 4Department of Scientific Research, Campus Ludes, Off-Campus Semmelweis University, 6912 Lugano, Switzerland; 5Castelnuovo della Daunia Thermal Medicine Center, Castelnuovo della Daunia, 71034 Foggia, Italy; 6Faculty of Medicine and Surgery, Vita-Salute San Raffaele University, 00166 Rome, Italy; 7Spasticity and Movement Disorders “ReSTaRt” Unit, Physical Medicine and Rehabilitation Section, OORR Hospital, University of Foggia, 71122 Foggia, Italy

**Keywords:** posture, Activities of Daily Living, physiotherapy, thermal care, rehabilitation

## Abstract

Background: The aim of this study was to evaluate the effects of integrated thermal rehabilitation care (ITRC) on postural balance and health-related quality of life in subjects with basic autonomy. Materials and Methods: From June to December 2021, a total of 50 individuals with six points on the Katz Index of Independence in Activities of Daily Living (ADL) and a mean age of 66 (DS ± 12), comprising 27 (54%) males and 23 (46%) females, were selected. This study was carried out at the Thermal Medical Center of Castelnuovo della Daunia (Foggia, Italy), which operates within the National Health Service. The outcome measures were baropodometry (static exam, dynamic exam, and stabilometric exam), a biometric evaluation system, and the EuroQol 5-Dimension (EQ-5D-5L). Results: Statistical analysis of the data showed how balance affected postural control and how ITRC was able to reduce the body’s imbalance and improve quality of life. The vertical angles in frontal projection displayed an increment in the values (head/shoulder, *p* = 0.009; head/pelvis, *p* = 0.001; right hip/knee, *p* = 0.01; right hip/ankle, *p* = 0.008). In a dynamic analysis, the podalic weight percentage was shown to have a reduction in imbalance on both sides (left side, *p* = 0.01; right side, *p* = 0. 01). EQ-5D-5L showed a statistically significant improvement in quality of life and perception of quality of life. Indeed, the health status score improved in all items and in the total rate of the EQ index. In all subjects, walking motility (*p* = 0.005), self-care (*p* = 0.002), and habitual activity (*p* = 0.002) showed statistically significant increments in their values. Pain/discomfort (*p* = 0.001) and anxiety (*p* = 0.006) were also reduced. In addition, there was a statistically significant increment in the Visual Analogue Scale (VAS) score (*p* = 0.001) for life perception. Conclusions: The ITRC approach showed how small adjustments and postural rebalancing led to a significant improvement in quality of life. ITRC can be considered an effective treatment with good tolerability for a variety of musculoskeletal disorders.

## 1. Introduction

Thermal therapies have long been used to improve the well-being of individuals. Traditional thermal approaches include mud therapy and baths in thermal waters with various mineral salts, such as sulfur, bicarbonate, sodium chloride, and bicarbonate–chloride [1]. The effects of thermal care are diverse; for instance, it provides benefits for the management of painful symptom, functional skills, and quality of life [2]. Furthermore, thermal water’s properties are well known in various fields, such as the treatment of breast cancer survivors. In that group of patients, thermal care was effective in reducing neck pain, shoulder pain, and muscular trigger points [2]. Traditional thermal therapies could be combined with other treatments, such as rehabilitation therapies. Indeed, the effects of thermal care and rehabilitation have been investigated for their provision of new opportunities for treating people with musculoskeletal disorders. Furthermore, a thermal therapy environment offers a wide range of rehabilitative interventions, such as physical therapy, aquatic therapies, and neuromuscular manual therapy integrated with pharmacological treatment [3]. In addition, manual neuromuscular therapy combined with thermal care showed beneficial effects in subjects who received it by activating visceral–somatic pathways and positively affecting the well-being of individuals [4]. The relationship between physiotherapy and environments of thermal therapy has been noted, but there is a lack of evidence of the effects of that combination on postural balance [5]. This is particularly true in persons with basic autonomy. Indeed, good posture depends on good support, so appropriate foot functions are crucial for achieving and maintaining a symmetric posture, which is evident in plantar pressure [6]. Certainly, correct biomechanics of the feet could have an important effect on postural balance in the standing position and during walking. Postural balance alterations are expressions of myofascial, visceral, and emotional disorders [7]. In the maintenance of work–life balance, biomechanical postural evaluation is crucial for assessing and managing dysfunctional disorders that came from different origins. Proper posture comes from the proper interconnection of different systems. A deficit in one of them could affect the balance of the entire body. Consequently, to restore equilibrium, a multidisciplinary approach could be necessary [8].

To measure the effects of good support on postural balance, in this study, we used biomechanical postural evaluation to evaluate individuals. This study aimed to determine the effectiveness of thermal care in combination with physiotherapy in individuals with a basic level of independence [7,9]. We hypothesize that good balance is fundamental for achieving and maintaining autonomy and that autonomy is crucial for living independently and improving quality of life [10]. Multimodal approaches could determine postural imbalance, especially in subjects with basic autonomy. Overall, the aim of this study was to evaluate effects of integrated thermal rehabilitation care (ITRC) on postural balance and, consequently, health-related quality of life in subjects with basic autonomy. Moreover, we would like to assess integrated approaches as a new way to treat individuals with reduced autonomy due to specific pathological pathways [8,11].

## 2. Materials and Methods

### 2.1. Study Design

This is a pilot retrospective observational study that was conducted to evaluate the effects of integrated thermal rehabilitation care on postural balance and, consequently, health-related quality of life in subjects with basic autonomy. From June to December 2021, a total of 50 individuals with 6 points on the Katz Index of Independence in Activities of Daily Living (ADL) were selected [12,13]. From Monday to Friday, 20 daily sessions of ITRC were performed. This study was carried out at the Thermal Medical Center of Castelnuovo della Daunia (Foggia, Italy). The ITRC protocol was composed of the following: manual neuromuscular therapy and underwater exercises in a specific warm pool (31 °C); respiratory exercises performed in a vascular path; mud at variable temperatures, followed by a hot thermal bath. Finally, all subjects underwent daily aerosol therapy in combination with manual therapy. The outcome measures were baropodometry (static exam, dynamic exam, and stabilometric exam), the results of a biometric evaluation system, and the 5-Dimension EuroQol (EQ-5D-5L). The data collection formed part of the professional teams’ routines, and participants were frequently assessed throughout the season. Therefore, the usual clearance from an ethics committee was not required. 

From June to December 2021, a total of 50 participants with 6 points on the Katz Index of Independence in Activities of Daily Living (ADL) [12,13] and a mean age of 66 (DS ± 12), comprising 27 (54%) males and 23 (46%) females, were selected. This study was carried out at the Thermal Medical Center of Castelnuovo della Daunia (Foggia, Italy), which operates within the National Health Service (Figure 1). This center had an agreement with the Department of Medical and Surgical Sciences of the University of Foggia (Italy) and Gemelli-Molise SpA (Italy) as well. This study was developed by following the Good Clinical Practice (GCP) guidelines. Written informed consent was obtained at baseline from all participants. 

The inclusion criteria were as follows: An age range of 40–70 years;Six points on the Katz Index of Independence in Activities of Daily Living (ADL);No critical cognitive impairments or compromised immune functions;Reading and understanding in patients’ own native language.

The exclusion criteria were as follows:History of cardiac, spinal, abdominal, or thoracic surgery in the past three years;Acute neurological and pulmonary disorders;Hypertension under uncontrolled drug treatment;Neoplasms in progress.

### 2.2. Outcome Measures

All patients were assessed (T0) before the protocol and at the end of all sessions (T1).

Baropodometry (Diasu Health Technologies—Rome, Italy) involved the assessment of plantar pressure on a platform. The system was composed of devices connected to software to analyze the pressures developed at different points in the plantar region for each foot. The baropodometric analysis included three types of analyses: a static exam, dynamic exam, and stabilometric exam. The static analysis recorded the pressure distributed on both feet and their percentage of loading. The dynamic exam analyzed the gait and the kinematic characteristics of walking while a person was walking along a platform from one end to the other. Clinical stabilometry was assessed by utilizing a high-resolution sensor to ensure surface and loading acquisition from a force platform. Through the surface and loading acquisition in the clinical stabilometry exam, the main parameters of postural adjustments and controls were analyzed. The stabilometric tests consisted of the assessment of the trajectory created by applying points of pressure on the body while the subject maintained a standing position. The neuromuscular control of the standing position involved vision, muscle–skeletal proprioception, and all components of the central nervous system. Standard tests with closed eyes and with open eyes were carried out after the registration of the patients’ personal data [14,15].Skeletal alignment was performed by using a playing platform combined with an Xbox console. An RGB–infrared camera was used with a Microsoft Kinect^®^ to grasp 3D frames. The software was able to create a 3D avatar composed of two projections (frontal and sagittal) and 20 anatomical landmarks. Postural evaluation was realized in different planes: frontal (anterior and posterior), sagittal (right and left lateral), and transverse (from above). A biometric evaluation system created a specific map of the body being examined with a 3D reconstruction of the skeleton and the spine. Indicators such as the pelvis, shoulders, spine, and foot were used to quantify structural misalignments in the body [13,16]. Furthermore, a dermatomal-level skin impedance assessment tool (ElectroNeuroFeedback (ENF), Carpenedolo, Brescia, Italy) was used to identify the somatic sites with the main dysfunctions. Considering skeletal alignment and results of stabilometry, it was possible to identify the myofascial structures with dysfunctions. Those areas received integrated somatic treatment in a thermal environment [17].EQ-5D-5L, which was introduced by the EuroQol in 1990, is the most widely generic tool for measuring health-related quality of life. It is applicable to a large range of health conditions and therapies, is understandable, and takes just a few minutes to complete. The questionnaire included five levels of severity in each of the five dimensions and the EQ VAS life score. The five dimensions were mobility, self-care, usual activities, pain/discomfort, and anxiety/depression, with five levels of severity (no problems, slight problems, moderate problems, severe problems, and extreme problems) [18,19].

### 2.3. Rehabilitation Protocol 

From Monday to Friday, a total of 20 daily sessions (1 h and 30 min) of ITRC were performed. The protocol was composed of manual neuromuscular therapy and underwater exercises in a specific warm pool (31 °C) for 30 min. During the underwater exercises, subjects were encouraged to walk with arm and leg coordination (10 min), to lift their heels (both legs three times for 10 × 3 sets), and to lift the heel of a single leg (10 × 3 sets three times) with proprioceptive floating boards. The group received manual therapy to deactivate the trigger points identified during the biophysical metric evaluation process. During manual stimulation, the physiotherapist applied pressure with the hands on localized trigger points of the patients. Subsequently, 10 min of respiratory exercises were performed in a vascular path with two thermal water tanks containing water for ozone therapy at 24 °C and 38 °C and equipped with jets at a pressure of 4–6 atmospheres. The breathing exercises consisted of deep breathing (10 × 2 sets three times) and deep breathing with spine mobilization (10 × 2 sets three times). Then, the patients underwent 20 min of mud therapy at various temperatures (40–42 °C); this was applied in the main areas of dysfunction that were identified during the evaluation (key trigger points), followed by a hot thermal bath (36–37 °C) for 12 min. Finally, all subjects performed daily aerosol therapy in combination with manual treatment of the pectoralis minor, elevator scapulae, and diaphragm muscles for 15 min. All subjects completed the protocol without adverse events.

### 2.4. Statistical Analysis

Statistical analysis was performed by using the NCSS© 9 statistical software package for Windows (NCSS© LLC, Kaysville, UT, USA). The data were expressed as the mean ± standard deviation (SD). The normality of the data was evaluated with the Shapiro–Wilk test, and it was found that the data were normally distributed. Accordingly, the analysis of the data before and after the intervention was performed with Student’s *t*-test (paired sample), which is a parametric test. Differences were compared at two time points—after (T1) versus before (T0) the therapeutic intervention. *p*-values of less than 0.05 were considered significant.

## 3. Results

The stabilometric evaluation (Table 1) showed an elliptical area in both standard tests, a decrease in the amplitude of the oscillations, and, consequently, an improvement in postural control (not statistically significant: *p* = 0.1). Although the results were not significant, the sway path and the standard test with opened eyes showed an increase in the postural tone compared to the standard test with closed eyes. 

The Romberg test showed an increase in its index in T1 (statistically insignificant). In the static analysis of the baropodometric profile (Table 2), the results demonstrated a reduction in the values of the percentage of the podalic weight and the podalic angle. Thus, there was a better body symmetry between the right and left sides (statistically insignificant). In the dynamic analysis, according to the percentage of podalic weight, a reduction in imbalance was shown on both sides (statistically significant).

In the postural evaluation (Table 3), in all measurements of the frontal projection, the horizontal angles showed a reduction in the values of the postural adjustments (statistically insignificant). The vertical angles in the frontal projection displayed an increase in the values (statistically significant). There was also an improvement in the values for the left hip/ankle and left hip/knee (statistically insignificant). In the evaluation of the skeletal symmetry of the head/pelvis, there was a decrease in the values, which meant that there was better postural control (statistically significant). In the sagittal projection, the values of the vertical angles remained stable.

The EQ-5D-5L questionnaire (Table 4) showed a reduction in all items and in the total EQ-Index (statistically significant). In addition, there was a statistically significant increase in the VAS score for life perception.

## 4. Discussion

Therapies in thermal environments increase circulation and relaxation and reduce muscular tension and pain in different clinical conditions, such as heart failure, chronic venous disease, and other chronic diseases. Thermal therapies can be applied in different contexts depending on the needs of the subjects. In addition, they can enhance general well-being and, consequently, quality of life [20]. In support of the literature that was analyzed, our results showed that multimodal rehabilitative interventions in addition to classical thermal therapy improved quality of life in terms of motion, emotional reactions, pain, and perception of quality of life. As a matter of fact, the health status score according to EQ-5D-5L improved in all items and in the total rate of the EQ-Index. In all subjects, walking motility, self-care, and habitual activity showed a statistically significant increase in performance. Pain/discomfort and anxiety were also reduced, which was probably because in the group, the therapy increased self-awareness and improved body flexibility. As a result, perception of life according to the VAS increased, and individuals felt more compliant with themselves and their mobility. In addition, Masiero has investigated the clinical effects of physical and thermal therapies in the form of non-invasive techniques that reduced pain symptoms and improved the well-being of individuals [21]. By adding thermal care and physiotherapy, the present study demonstrated how this combination could increase body relaxation and, consequently, achieve better body symmetry. In fact, in the dynamic analysis, the podalic weight percentage demonstrated statistically significant maintenance of symmetry on the right side compared to the left side. In addition, postural control was related to complex mechanisms that should work together to ensure balance in the body. In particular, in frontal projection, the protocol achieved a reduction in muscle tension and an increase in flexibility to approach symmetry on both sides. Thermal therapies can be applied in different contexts, one of which could be an aquatic environment. Indeed, the properties of underwater therapies have been noted, and thermal aquatic exercises provide clinical benefits in terms of edema control, range of motion, and relaxation [22,23]. Our protocol included manual neuromuscular therapy and water exercises in a specific warm pool (31 °C). Indeed, physiotherapy in aquatic and warm environments reduced the loading of gravity on the joints and improved the relaxation of the body. Consequently, it could positively influence the sensory and motor reorganization of the central nervous system. Therefore, the physical characteristics of the water during physical activities could stimulate the acquisition of new neuro–motor–sensory skills that have been impaired due to pathologies [24]. That study took advantage of all properties of thermal water, and in a specific warm pool, we created a training exercise that we included in the protocol. As we have mentioned, the thermal approach is well known for its antalgic, muscle-relaxant and -trophic, anti-edemigenous, and anti-inflammatory properties. It could be combined with rehabilitation to provide well-being and to prevent and treat neuromuscular impairments [20]. In addition, the literature strongly supports the immune-beneficial role of thermal therapy in patients with localized and advanced cancer [25]. According to the literature showing the wide spectrum of its beneficial effects, we believe in the clinical efficacy of thermal therapy that includes physical therapy. To our knowledge, this is the first report to evaluate this combination and its effects on postural control. Postural balance is the result of the interaction of different systems that should work together to guarantee equilibrium during movements and during relaxation. The connection between the postural system and balance of the body is related to feedback and feedforward postural receptors, such as podals and oculars. From the body’s periphery, the nervous system generates a postural muscular adjustment to achieve the body’s center of gravity and, consequently, balance [26]. Bodily adjustments lead to specific changes for achieving continuous and spontaneous balance of the body in space [27]. Due to the multitude of components of postural control, our result supports the efficacy of multimodal therapies for balance of the body. As expected, in the stabilometric evaluation, the results of the elliptical area showed a reduction in the amplitude of the oscillations and, consequently, an improvement in postural control. In addition, the results of the oscillations (sway) in the standard closed-eye test (where only musculoskeletal proprioception was active) and in the standard open-eye test (where visual perception was added to proprioception) showed better motor control with open eyes in comparison with that with closed eyes. The open-eyes standard test showed an increase in postural tone compared to that in the standard test with closed eyes. The Romberg index generated by the Romberg test made it possible to evaluate the risk of falling. The Romberg index confirmed a reduction in proprioception that was functionally related to a reduction in postural balance and vestibular contribution. In the static analysis, the values of the foot weight percent and podalic angle indicated that from T0 to T1, the foot pressure on the footplate was more symmetrical between the two sides. Thus, better symmetry led to better stability and, consequently, more postural control. Although the data were not statistically significant, they showed a normalization of the static and dynamic balance. Furthermore, in the dynamic analysis, the podalic weight percentage demonstrated statistically significant maintenance of symmetry on right side in comparison with the other side. In the frontal view, the horizontal angles showed that there was a reduction in shoulder symmetry at T1. Instead, the symmetry of the pelvis and knees improved by approaching 0°. The results for the frontal plane demonstrated significant dimorphisms in the vertical angles of the line passing along the midpoint of the body and anatomical landmarks of the head/shoulder, right hip/knee, and right hip/ankle. The vertical angles in the frontal view of the left hip/ankle and left hip/knee maintained a state of equilibrium. Finally, the vertical head/pelvis angles showed a statistically significant alignment of the body. In the sagittal view, the values of the vertical angles showed that the vertebral column dimorphisms in the frontal plane did not change from T0 to T1 for all parameters examined. Overall, the data showed how ITRC could reduce body imbalances and improve quality of life. In subjects with a good rate of basic autonomy, we found that ITRC in combination with the Biophysics Metric pathway increased the ability to manage daily life and self-care. We hypothesized that by improving the perception of postural balance, ITRC could decrease discomfort, anxiety, and, consequently, painful symptoms. Physiotherapy combined with the effects of a thermal environment could allow the introduction of a new model of care for individuals with basic autonomy, but it could be also a new way to evaluate its efficacy in other clinical contexts. 

### Weaknesses and Strengths of This Study

Weaknesses: The first limitation of this study involved the characteristics of the subjects who were enrolled. A description of the population could allow knowledge of the participants to which the results of this study can be confidently applied. The second limitation was the lack of an extended follow-up to observe the effectiveness of the ITRC protocol. Thirdly, enrollment could be stratified on the basis of subjects’ subgroup characteristics. If the experience of particular subgroups of subjects was important and likely to vary, then enrollment should be stratified on the basis of those subjects’ subgroup characteristics. Finally, another limitation of this study was the relatively small sample size, which was not sufficient for finding (assessing) differences in parameter estimations; therefore, our study could have low external validity.

Strengths: Patients’ satisfaction during the treatment was great. The timing and the relationship with the health professionals probably increased their curiosity and motivation for accessing the protocol. This study reported the statistical significance of data and the causes and effects of the relationship between exposure and outcomes. This study observed the effects of ITRC on postural balance and, consequently, health-related quality of life in subjects with basic autonomy. It could represent a new vision of cooperation among thermal care, rehabilitation, and postural balance, which are crucial for living autonomously. Further study will be necessary to compare this protocol with a population with a low rate of autonomy and specific diseases, contexts, or therapies. 

## 5. Conclusions

The ITRC approach showed how small adjustments and postural rebalancing led to a significant improvement in the quality of life. ITRC can be considered an effective treatment with good tolerability and no adverse events. In a thermal environment, the Biophysics Metric pathway could be a non-invasive approach that can reduce discomfort and improve the quality of life in subjects with basic autonomy. Further studies are necessary to define the protocol and to detect the effects of ITRC on subjects with different levels of autonomy or other pathologies.

## Figures and Tables

**Figure 1 healthcare-11-02384-f001:**
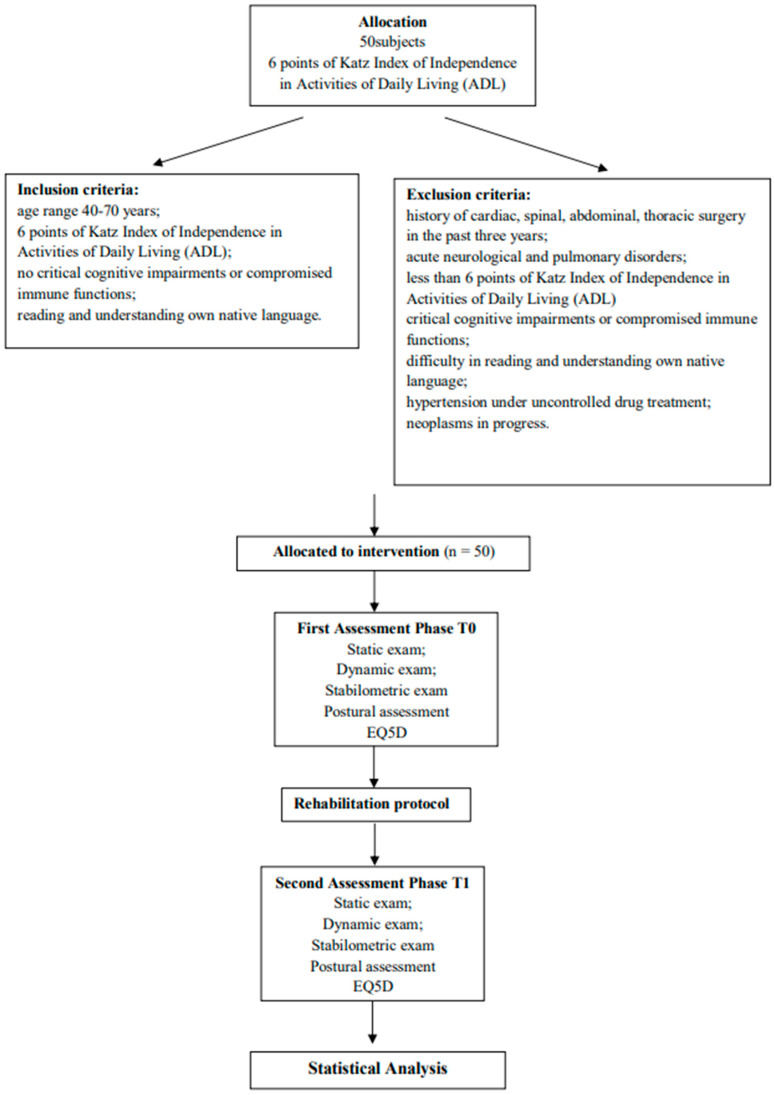
Design and flow of this study.

**Table 1 healthcare-11-02384-t001:** Comparison of pre-intervention and post-intervention stabilometric values.

Variable	Count	Mean ± SD	** p*
ELLIPSE AREA (mm^2^) (T0) OPENED EYES	50	4 ± 2	
ELLIPSE AREA (mm^2^) (T1) OPENED EYES	50	3 ± 2	0.1
ELLIPSE AREA (mm^2^) (T0) CLOSED EYES	50	6 ± 6	
ELLIPSE AREA (mm^2^) (T1) CLOSED EYES	50	5 ± 4	0.2
SWAY PATH (mm/s) (T0) OPENED EYES	50	225 ± 139	
SWAY PATH (mm/s) (T1) OPENED EYES	50	199 ± 119	0.1
SWAY PATH (mm/s) (T0) CLOSED EYES	50	313 ± 309	
SWAY PATH (mm/s) (T1) CLOSED EYES	50	275 ± 222	0.1
ROMBERG INDEX (T0)	50	166 ± 208	
ROMBERG INDEX (T1)	50	200 ± 240	0.3

* The *p*-value shows differences between T0 and T1 for the group (*t*-test). The cut-off for statistical significance of the *p*-value is ≤0.05.

**Table 2 healthcare-11-02384-t002:** Comparison of pre-intervention and post-intervention baropodometric values.

Variable	Count	Mean ± SD	*p*
% PODALIC WEIGHT T0 left (Static Analysis)	50	44 ± 9	
% PODALIC WEIGHT T1 left (Static Analysis)	50	46 ± 9	0.1
% PODALIC WEIGHT T0 right (Static Analysis)	50	53 ± 10	
% PODALIC WEIGHT T1 right (Static Analysis)	50	51 ± 10	0.1
PODALIC ANGLE T0 left (Static Analysis)	50	10 ± 5	
PODALIC ANGLE T1 left (Static Analysis)	50	9 ± 5	0.3
PODALIC ANGLE T0 right (Static Analysis)	50	12 ± 6	
PODALIC ANGLE T1 tight (Static Analysis)	50	12 ± 5	0.9
% PODALIC WEIGHT T0 left (Dynamic Analysis)	50	43 ± 19	
% PODALIC WEIGHT T1 left (Dynamic Analysis)	50	39 ± 18	* 0.01
% PODALIC WEIGHT T0 right (Dynamic Analysis)	50	42 ± 18	
% PODALIC WEIGHT T1 tight (Dynamic Analysis)	50	46 ± 20	* 0.01
PODALIC ANGLE T0 left (Dynamic Analysis)	50	11 ± 6	
PODALIC ANGLE T1 left (Dynamic Analysis)	50	10 ± 6	0.1
PODALIC ANGLE T0 right (Dynamic Analysis)	50	11 ± 6	
PODALIC ANGLE T1 tight (Dynamic Analysis)	50	11 ± 6	0.5

* The *p*-value shows differences between T0 and T1 for the group (*t*-test). The cut-off for statistical significance of the *p*-value is ≤0.05.

**Table 3 healthcare-11-02384-t003:** Comparison of the pre-intervention and post-intervention postural evaluation values.

Variable	Count	Mean ± SD	*p*
*Frontal projection (horizontal angles)*			
(T0) SHOULDER	50	0.1 ± 1	
(T1) SHOULDER	50	0.3 ± 1	0.08
(T0) PELVIS	50	0.5 ± 1	
(T1) PELVIS	50	0.1 ± 1	0.05
(T0) KNEE	50	2 ± 4	
(T1) KNEE	50	1 ± 4	0.05
*Frontal projection (vertical angles)*			
(T0) HEAD/SHOULDER	50	0.7 ± 3	
(T1) HEAD/SHOULDER	50	0.8 ± 3	* 0.009
(T0) HEAD/PELVIS	50	0.5 ± 1	
(T1) HEAD/PELVIS	50	0.2 ± 1	* 0.001
(T0) HIP/KNEE LEFT	50	1.5 ± 3	
(T1) HIP/KNEE LEFT	50	1.9 ± 3	0.4
(T0) HIP/KNEE RIGHT	50	0.3 ± 3	
(T1) HIP/KNEE RIGHT	50	0.8 ± 3	* 0.01
(T0) HIP/ANKLE LEFT	50	1.1 ± 2	
(T1) HIP/ANKLE LEFT	50	1.1 ± 2	0.9
(T0) HIP/ANKLE RIGHT	50	0.04 ± 1	
(T1) HIP/ANKLE RIGHT	50	1 ± 2	* 0.008
*Sagittal projection (vertical angles)*			
(T0) HEAD/SHOULDER	50	13 ± 6	
(T1) HEAD/SHOULDER	50	13 ± 6	0.6
(T0) HEAD/PELVIS	50	5 ± 3	
(T1) HEAD/PELVIS	50	5 ± 4	0.4
(T0) HIP/KNEE LEFT	50	16 ± 3	
(T1) HIP/KNEE LEFT	50	16 ± 3	0.2
(T0) HIP/KNEE RIGHT	50	16 ± 3	
(T1) HIP/KNEE RIGHT	50	16 ± 3	0.2
(T0) HIP/ANKLE LEFT	50	13 ± 2	
(T1) HIP/ANKLE LEFT	50	14 ± 3	0.2
(T0) HIP/ANKLE RIGHT	50	14 ± 2	
(T1) HIP/ANKLE RIGHT	50	14 ± 2	0.3

Frontal projection (vertical angles); sagittal projection (vertical angles): planes from skeletal alignment performed by using an RGB–infrared camera abled to grasp 3D frames. * *p*-values show differences between T0 and T1 for the group (*t*-test). The cut-off for statistical significance of the *p*-value is ≤0.05.

**Table 4 healthcare-11-02384-t004:** Comparison of the pre-intervention and post-intervention EQ-5D-5L values.

Variable	Count	Mean ± SD	*p*
MOTILITY (T0)	50	0.05 ± 0.09	
MOTILITY (T1)	50	0.005 ± 0.02	* 0.005
SELF-CARE (T0)	50	0.04 ± 0.06	
SELF-CARE (T1)	50	0.01 ± 0.03	* 0.002
USUAL ACTIVITY (T0)	50	0.01 ± 0.02	
USUAL ACTIVITY (T1)	50	0.005 ± 0.01	* 0.002
PAIN/DISCONFORMT (T0)	50	0.1 ± 0.1	
PAIN/DISCONFORMT (T1)	50	0.04 ± 0.05	* 0.001
ANXIETY/DEPRESSION (T0)	50	0.04 ± 0.06	
ANXIETY/DEPRESSION (T1)	50	0.01 ± 0.02	*0.006
EQ-Index (T0)	50	0.3 ± 0.6	
EQ-Index (T1)	50	0.8 ± 0.8	* 0.001
VAS (T0)	50	62	
VAS (T1)	50	81	* 0.001

* *p*-values show differences between T0 and T1 for the group (*t*-test). VAS: Visual Analogue Scale. EQ-Index: EuroQol Index. The cut-off for statistical significance of the *p*-value is ≤0.05.

## Data Availability

Data sharing not applicable.

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
