# Peer review of "Integrated Thermal Rehabilitation Care: An Intervention Study"

_healthcare, 2023, doi:10.3390/healthcare11172384_

Round 1
Reviewer 1 Report
healthcare-2357723
Thank you very much for inviting me to revier the paper entitled: Integrated Rehabilitation Thermal Care: an intervention study
General comments: interesting manuscript that addresses an important área of medical need. The authors should give answers to important questions about effects of Integrated Rehabilitation Thermal Care (IRCT) on postural balance and consequently Health-related quality of life in subjects with basic autonomy.
Please see may specific comments below for more details.
This manuscript consists of a non-structured abstract with 5 keywords, 5 sections (introduction, methods with 4 subsections (subsection 2.2 with 3 sections), results, discussion with 1 subsection, and conclusions) on 13 pages of single-spaced text with embedded figures and tables. There are 27 references,1 figure and 4 tables.
Specific comments:
1. The keywords are absolutely fine.
2. Background: too much information. Restructure part of Material and methods and Results.
3. Introduction: Missing references (Lines 49-51; 53-58; 63-68…the authors only used 10 references in it). Also, a single paragraph, unstructured information. Where is noted the relationship between physiotherapy and thermal environment? Line 65/66: “person with basic rate of autonomy???? The authors talk about specific pathological pathways and people with limited autonomy but do not specify anything.
4. Materials and Methods: If we are talking about patients with functional limitations and limited autonomy... What professional teams are they playing on?. It is not clear where they have taken the sample of patients. Although the data collection is of the professional teams'routines, it is not clear to me that the approval of the ethics committee is not necessary. All patients in any hospital or Health center are evaluated frequently, and informed consent as well as the Ethics Committee are necessary to be able to disclose and work with these data.
In addition, Figure 1 is not readable. It is possible to add images of the measurements using Xbox console, Baropodometry platform... Stabilometric tests...
Why this rehabilitation protocol and not another? What kind of pathologies do patients present?
5. Discussion: As in introduction, many sentences affirming benefits that are not referenced (Lines 210.211.212…) Line 214 “In support of literatures analyzed…” which one?? No references. Line 223 “several studies…” the authors only add 1 reference from 2008 and it is not a Review or Meta-analysis where the results of several studies are concluded. Again the discussion is in a single paragraph difficult to read. They must restructure and reference their claims.
6. Weaknesses and Strengths of the Study: As the authors explain in their limitations/weaknesses, if you do not have a homogeneous group of patients... It is difficult to conduct this study.
Thanks again for the invitation.
Author Response
We appreciate a lot this reviewer’s suggestion
- Introduction:Missing references (Lines 49-51; 53-58; 63-68…the authors only used 10 references in it). Also, a single paragraph, unstructured information.
As we have mentioned: “The relationship between physiotherapy and thermal environment is noted but, there is lack of evidence on effects of that combination on postural balance.” We analyzed evidences so close with our hypothesis and there are enough to justify our proposals.
- Where is noted the relationship between physiotherapy and thermal environment? It was well described
Traditional thermal therapies could be combining with others treatments such as rehabilitation therapies. Indeed, effects of thermal care and rehabilitation have been investigated as new opportunity in people with musculoskeletal disorders. Besides, thermal environment offers a wide range of rehabilitative interventions such as physical therapy, aquatic therapies or neuro-muscular manual therapy integrated with pharmacological treatment as well [2, 3]. In addition, neuro-muscular manual therapy combined with thermal care showed beneficial effects in subjects who received that therapy, activating visceral-somatic pathways and affecting positively wellbeing of individuals [4].
- Line 65/66: “person with basic rate of autonomy???? The authors talk about specific pathological pathways and people with limited autonomy but do not specify anything.
We didn’t talk about specific pathological pathways and people with limited autonomy because we evaluated subjects with a basic rate of autonomy independently from specific pathological pathways. Basic rate of autonomy means 6 points of Katz Index of Independence in Activities of Daily Living (ADL). This rate is referred to the basic and everyday skills that are essential to living independently.
- Materials and Methods: If we are talking about patients with functional limitations and limited autonomy... What professional teams are they playing on?. It is not clear where they have taken the sample of patients. Although the data collection is of the professional teams' routines, it is not clear to me that the approval of the ethics committee is not necessary.All patients in any hospital or Health center are evaluated frequently, and informed consent as well as the Ethics Committee are necessary to be able to disclose and work with these data.
We didn’t talk about specific pathological pathways and people with limited autonomy because we evaluated subjects with a basic rate of autonomy independently from specific pathological pathways.
- The data collection formed part of the professional teams’ routines in which players are frequently assessed across the season. Therefore, the normal ethics committee clearance was not required.
We couldn’t analyze in that study patient’s clinical folder. The study was developed following the Good Clinical Practice (GCP) guidelines. It was conducted within the ethical principles outlined in the Declaration of Helsinki, and with the procedures defined by the ISO 9001-2015 standards for “Research and experimentation”. Written informed consent was obtained at baseline from all participants. (REF Winter, EM and Maughan, RJ. Requirements for ethics approvals. J Sports Sci 27: 985, 2009)
- In addition, Figure 1 is not readable.
I have attached PDF of the flow chart during submission so the reviewer can look at it properly
- It is possible to add images of the measurements using Xbox console, Baropodometry platform... Stabilometric tests...
Sorry, is not possible
- Why this rehabilitation protocol and not another? What kind of pathologies do patients present?
Because we created a new way of multimodal approach and we have tested it in that group
- Discussion: As in introduction, many sentences affirming benefits that are not referenced (Lines 210.211.212…) Line 214 “In support of literatures analyzed…” which one?? No references. Line 223 “several studies…” the authors only add 1 reference from 2008 and it is not a Review or Meta-analysis where the results of several studies are concluded. Again the discussion is in a single paragraph difficult to read. They must restructure and reference their claims
The discussion section is parts of our research where we described, analyzed, and interpreted our findings. The study explained the significance of results and tied and inferred everything back to the research questions. References have been analyzed and mentioned in the body of the text in the square brackets.
- Weaknesses and Strengths of the Study: As the authors explain in their limitations/weaknesses, if you do not have a homogeneous group of patients... It is difficult to conduct this study.
Indeed we have described it as Weaknesses point. This study has helped us to evaluate the effects of new way of therapeutic approach. Further study with a homogeneous group of patients will help to understand the effects in specific contexts. Unfortunately, we did not execute a sample size study, since, at best of our knowledge, this is the first study dealing with the beneficial role of IRCT and posture, and was difficult to postulate adequate confidence interval to calculate sample. In the limitation of the study we also declare that our study could be biased by the lack of a sample size study, with a low external validity.

Reviewer 2 Report
Preliminary studies of patients undergoing IRCT therapy have shown that we should continue to look for effective methods to improve the quality of life of patients with pain and balance problems.The work is coherent, the research problem is thought out and well developed. The topic of the work is current.
In the opinion of the reviewer, the article can be accepted for publication.

Author Response
We appreciate a lot this reviewer’s suggestion

Reviewer 3 Report
Dear authors, it seems to me a well-executed study, although with a small sample. But why in the discussion and in the conclusions do they talk about statistical significance and state that it is when p>0.05?
Author Response
We appreciate a lot this reviewer’s suggestion
- Dear authors, it seems to me a well-executed study, although with a small sample. But why in the discussion and in the conclusions do they talk about statistical significance and state that it is when p>0.05?
Just to recognize the significant value. Is has been removed
Reviewer 4 Report
Need more details on methods manual therapy.
Discussion is lack of analying the results with previous studies.
why and how those data with significance and not must be discussed.

minor mitakes
Author Response
We appreciate a lot this reviewer’s suggestion
- Discussion is lack of analying the results with previous studies. why and how those data with significance and not must be discussed.
The discussion section is parts of our research where we described, analyzed, and interpreted our findings. The study explained the significance of results and tied and inferred everything back to the research questions. References have been analyzed and mentioned in the body of the text in the square brackets. This is the first study dealing with the beneficial role of IRCT and posture and the background of references helped us to understand the importance of each part of the protocol and finally our therapeutic proposal.
Round 2
Reviewer 1 Report
It is true that research on water treatment is needed. The authors have not modified neither Introduction nor Discussion, for them it is correct, personally I continue to see it with few references and without structure. I think the abstract is too long... I cannot accept the manuscript for publication in the current form.
Author Response
Thank you for your suggestion. Your corrections have been executed.

Reviewer 4 Report
The diagram is not visible.

Author Response

(The authors gave the same response as above.)
